# A Comparison of Robust Parsing Methods for HPSG

## Abstract

This paper explores several techniques for enhancing coverage when parsing with HPSG grammars, determines appropriate evaluation methods, and uses them to compare performance. Depending on the dataset, baseline coverage gaps can be reduced by between 75% and 100%, while simultaneously improving EDM F1 scores.

## 1 Introduction

Head-driven Phrase Structure Grammar (Pollard and Sag, 1994) is a constraint-based grammar formalism that combines linguistically motivated descriptive mechanisms with precise and computationally tractable logical and combinatorial properties. The design goals for computational language systems generally include the following (doubtless in addition to many others):

1. coverage — *all* naturally-occurring inputs can be analyzed
2. precision — *only* naturally-occurring inputs can be analyzed[1]
3. efficiency — analysis happens quickly
4. semantic competence — meaning[2] is preserved and made explicit
5. succinctness — knowledge embodied by the system is well-organized

The relative priority of these goals tends to differ from one system to the next, however. Whereas many system designers will rank coverage and perhaps efficiency as the most important of these five desiderata, most HPSG grammarians would rank precision, semantic competence, and possibly succinctness higher (Flickinger, 2011).

---

[1]Or inputs that could reasonably be expected to occur in some natural context.

[2]By this is meant that portion of the meaning of an input which is of specific interest to the system in question.

While this is not the place to expound at length on the motivation for a precision grammar, it is perhaps instructive to mention two important benefits: first, by excluding analyses of strings that the grammarian judges to be outside of the language in question, the problem of *spurious ambiguity* is typically greatly reduced for strings that *are* grammatical. Second, precision is a prerequisite to bidirectionality, enabling both parsing and generation. A system that cannot differentiate between strings that are in a language and those that are not cannot be used in this way.

The relatively low priority assigned to coverage in many HPSG-based computational linguistics projects is a natural corollary of the premium placed on precision, together with the Zipfian distribution of language. However, given that precision grammars are typically hand-crafted resources embodying substantial amounts of human effort, restricted coverage can at times be quite frustrating. In particular, the inability of a precision grammar to produce any analysis whatsoever for some proportion of inputs means an application built on such a grammar has two options:

1. Ignore such inputs
2. Rely upon evidence from a second analysis component, resulting in a non-homogeneous data flow

Both options are viable in certain circumstances; however, neither option is very appealing. Having to depend on two different types of signal makes the design of downstream components more complicated, and depending on the rate of coverage gaps, ignoring unparseable inputs entirely could be disastrous. A better solution is needed.

This paper examines the issue of incomplete coverage in HPSG grammars, and explores several mitigation techniques, collectively dubbed *robust parsing*. Specifically, the English Resource Grammar (Flickinger, 2000) is augmented with

extended coverage by two techniques involving adding low-precision HPSG rules, and two techniques involving automatic constraint relaxation. The techniques are described in detail in Section 3, and the results are presented in Section 4. Finally, the effectiveness of the techniques is examined and commented on in Section 5.

A secondary contribution of this paper involves methodology for the evaluation of such robust parsing techniques, which is somewhat nonobvious due to the fact that gold standard evaluation data for the English Resource Grammar (henceforth ERG) is only readily available for incoverage inputs.

## 2 Related Work

The desire for increased coverage is at least as old as the field of computational linguistics. In the context of parsing with PCFGs, a time-honored solution is to assume that any two nonterminals can combine to form any other nonterminal (e.g. PP→VP+DET), but only with extremely low probability. This technique is referred to as *smoothing*. The result of smoothing is that any input string can be assigned a sentential analysis. A major drawback is the resulting explosion of ambiguity and processing cost: any binary tree with labels drawn from the nonterminals is a legal analysis for every input of the corresponding number of words. However, if the extra probability mass added when smoothing the grammar is distributed judiciously, and if suitable efficiency methods (such as pruning) are employed, it is often still possible to arrive at useful analyses in reasonable amounts of time.

The idea behind smoothing can, in principle, be applied to parsing with HPSG as well. Instead of having an enumerated set of nonterminal symbols, these grammars use *feature structures* to represent categories. The translation is roughly that any pair of adjacent sub-analyses with feature structures X and Y can license a larger analysis with an arbitrary feature structure Z. There are two major flies in this ointment. First, since HPSG does not posit a notion of probabilities,[3] there is no native way to document these licentious edges as suspect. Second, the inventory of possible feature structures Z to be licensed in each case is not necessarily bounded: while in some grammars the collection

of possible feature structures may be enumerable, it is likely to be very large, and in other grammars it may not even be finite, since recursive structures are possible. Nonetheless, this avenue of attack has been explored in some depth by Fouvry (2003b). Instead of licensing completely arbitrary feature structures Z, Fouvry gives a technique whereby existing grammar rules can be used even in circumstances where they formally cannot apply, to produce a more constrained inventory of "robust" subanalyses Z. Fouvry also addresses how to keep track of the degree of badness of these robust analyses, by associating to each of them a score indicating how many paths within daughter feature structures violated constraints on the rule. In a refinement, Fouvry (2003a) also shows how to use a more articulated score, where violating some constraints is considered more egregious than violating others.

Unfortunately, Fouvry does not give an empirical evaluation of his work. Indeed, he notes that the evaluation of robust parsing methods for HPSG is challenging, in part because gold standard evaluation data tends to be missing for inputs that cannot be parsed.[4] Even with Fouvry's careful limiting of the space of robust subanalyses, the search space is likely to increase to the point that parsing of longer sentences is unlikely to be practical within reasonable resource limits. However, Fouvry's work may deserve the title of the most principled approach to robust HPSG parsing.

A number of considerably less general but arguably more practical approaches have also been deployed. Among these is the unknown word handling mechanism used in the ERG (Adolphs et al., 2008). A part-of-speech tagger with its own robustness mechanisms (including access to a larger lexicon, a smoothed HMM, and suffix analysis) provides guesses about the coarse-grained part of speech of each word. In cases where the ERG has no native lexical coverage for a word, the part of speech tags are used to trigger "generic" lexical entries. Although these generic lexical entries are somewhat underspecified relative to native entries in terms of information about the syntax and semantics of the word in question, their presence usually enables the grammar to find an analysis for sentences that would otherwise have been un-

---

[3]Rather, ranking of competing analyses of ambiguous sentences is considered a separate operation.

[4]This is because the gold standard treebanks are typically prepared by manually disambiguating parse forests prepared with the grammar's help (Oepen et al., 2004).

parseable.[5]

Another relatively common (although unglamorous) reason for failing to produce results in the ERG is resource exhaustion:[6] no result was found within the amount of memory or time the user allowed, even though a result might well have been found if more resources were available to devote to the search. Dridan (2013) presents an effective extension of a familiar technique called supertagging to reduce resource consumption: a trigram sequence tagger is used to discard unlikely lexical categories, resulting in (depending on the threshold used) vastly reduced search space. Dridan reports not only an increase in coverage resulting from this reduced resource exhaustion, but also an increase in EDM F1 score and speed.[7]

Sometimes parsing fails due to reasons related to the structure of the input, rather than missing lexical entries or resource limits. Flickinger and Packard (2015) present a technique they dub *bridging*, in which two highly generic rules are added to an HPSG grammar in such a way as to allow any two constituents to combine, but yielding a highly constrained result feature structure which cannot then combine with anything other than more of its own type. Their technique is one of the ones evaluated in this paper, and is discussed in more depth in Section 3.4.2.

Zhang and Krieger (2011) show how to use treebanks built using an HPSG grammar to induce a PCFG that can produce analyses which look superficially like HPSG derivations. With suitable choice of parameters, such a PCFG can be quite robust. Unfortunately, the PCFG-produced analyses are frequently not fully coherent with respect to the constraints stipulated in the original HPSG grammar, preventing their use for extracting semantic information. Zhang et al. (In prep) subsequently proposed a robust unification technique which, combined with Zhang and Krieger's parser, allows nearly-coherent semantic analyses to be built using the original HPSG grammar, guided by the PCFG-produced trees. This technique is also

---

[5]This mechanism is enabled in all of the experiments described in this paper.

[6]The rate of occurrence is highly dependent on sentence length. Allowing up to 1GB of memory and 60 seconds of time on Wikipedia text with a 2.6GHz Opteron 6344 CPU, resource exhaustion accounts for about 6% of inputs but more than 50% of parse failures.

[7]Dridan's mechanism is not used in our experiments, except with the `pacnv` system, where resource exhaustion without it was judged too excessive for useful results.

one of the ones evaluated in this paper, and is discussed in more depth in Section 3.4.4.

The next section will give more details about the components used, techniques evaluated, and experiments performed in this comparison of robust parsing methods.

## 3 Methods

### 3.1 English Resource Grammar

Development on the ERG began in 1994 at Stanford as a component of the German-English machine translation project Verbmobil, with the initial focus on parsing and generating relatively short English sentences and phrases in spoken dialogues about meeting scheduling. This HPSG grammar has been under continuous expansion since that rather modest beginning, and has been used in research and/or commercial applications in automated customer email response, online education for teaching English grammar as well as predicate logic, information extraction over the English Wikipedia, and shared tasks in simulated robot control and in grammatical error detection in scientific writing. Alongside the grammar, which can by now successfully analyze at least 9 in every 10 sentences across a wide variety of text domains (Flickinger, 2011), a manually constructed treebank of some 1.5 million words parsed by the ERG has been built both for regression testing and as training material for automatic parse selection. This resource, dubbed the Redwoods treebank (Oepen et al., 2004), stores for each parsed sentence both the correct syntactic derivation (if available in the parse forest) and the associated meaning representation expressed in Minimal Recursion Semantics, henceforth MRS (Copestake et al., 2005). Of the roughly 10% of sentences in an arbitrary English text that the ERG fails to parse, half typically fail due to the parser hitting resource limits (due to too much ambiguity present in exhaustive parsing), and half fail due to remaining linguistic shortcomings of the ERG for phenomena in the long tail of less frequent constructions in English.

### 3.2 Evaluation methodology

It seems clear that the underlying goal of robustness techniques like the ones explored herein is to provide a larger quantity of useful analyses to a downstream process than could previously be produced. To that end, the most interesting quantity to

evaluate would be improvement or regression on some collection of tasks for which the ERG's parsing capability is an important component, i.e. an *extrinsic* evaluation. To do a good job of extrinsic evaluation, however, two things are required: first, a large enough annotated dataset to acquire meaningful statistics for a particular task, and second, a sufficiently varied collection of tasks to exercise all interesting aspects of the analyses. Unfortunately such work is beyond the scope of this paper. A lesser attempt at extrinsic evaluation seems tempting, but prone to errors.

Therefore, in this work we content ourselves with *intrinsic* evaluations of robust parsing techniques. Intrinsic evaluation is not without its own difficulties. The primary problem is that the sentences of greatest interest for evaluation, i.e. the ones that the grammar cannot by itself produce an analysis for, are exactly the ones for which no gold standard annotations are available.[8] This is because the ERG's gold standard annotations are built by manually disambiguating the forest of (often millions of) analyses licensed by the grammar.

### 3.3 Datasets

There are two ways to solve this problem. First, the missing gold standard data can be fabricated by a labor-intensive process we playfully dub "alchemy". This involves making minor edits to the unparseable sentence and/or the grammar (for instance, declaring a new multi-word lexeme in the grammar, or fixing a typo in the input) in order to produce the desired analysis.[9] Once the desired "gold" analysis has been recorded, the edits to the sentence and grammar can be discarded. The `alchemy45` dataset used in our evaluations was produced this way, from a collection of Wall Street Journal sentences that the `1212` release of the ERG was unable to analyze within reasonable resource limits (8 of the 45 items were actually parseable with more generous resource limits; the rest were properly outside of the grammar's coverage). Our `ws13r` dataset was produced in the same way, from ERG-unparseable Wikipedia sentences, but in this case, relative to a more recent grammar version.

The second approach is to perform robust parsing with respect to a slightly out-of-date version of the ERG, and evaluate the results only against inputs that were out of scope in that version but are in scope in a more recent grammar version. Our `semcor` and `wsj00ab` datasets follow this methodology; they consist of items from the Brown Corpus (Kucera and Francis, 1967) and Wall Street Journal, respectively, that had no gold standard analysis in the `1214` release of the grammar, but do have gold standard analyses in the current SVN trunk version of the grammar. This approach is much cheaper in terms of manual labor. However, it has the disadvantage (shared with our `ws13r` dataset) that comparison between system results and the gold standard necessarily involves two distinct grammar versions, and hence differences can be expected even for a perfect parser. We attempt to minimize the effect of these differences for the specific grammar pair of `1214` vs `trunk` by taking the following measures.

1. Rather than only judging exact matches, we give partial credit for partial MRS match, using the Elementary Dependency Matching (EDM) evaluation metric presented by Zhang et al. (In prep).

2. Internal properties of semantic variables, such as number or tense, are not included in evaluation, since there were significant changes in how they are computed between `1214` and `trunk`.

3. Information-structure predicates (bearing the suffix `_d_rel`) are ignored, since these were relocated in `trunk`.

4. Only the lemma field within predicate names is compared, as many subsense fields were changed.

5. Some slippage is allowed in aligning character positions recorded within each MRS.

The datasets used for evaluation are as follows:

| Dataset | Items | $\frac{\#words}{\#items}$ | In 1212? | Version |
|---|---|---|---|---|
| `alchemy45` | 45 | 28.5 | no parse | `1212` |
| `ws13r` | 207 | 27.7 | no parse | `trunk` |
| `semcor` | 241 | 24.8 | no gold[10] | `trunk` |
| `wsj00ab` | 76 | 25.6 | no gold | `trunk` |

### 3.4 Parsing systems

We compared the performance of five approaches to increasing the robustness of parsing with the ERG. These techniques are described below.

---

[8] In fact, they are a subset; there are some inputs that the grammar is capable of analyzing, but not with the intended interpretation.

[9] It would also of course be possible to manually type out the desired analysis, but this is even more labor intensive and extremely error-prone, given the degree of connectivity and detail recorded in MRS.

[10] Some `semcor` and `wsj00ab` items are parseable under `1214`, but not with the intended analysis.

### 3.4.1 Baseline

The ERG by itself cannot, by definition, provide analyses for any inputs that it considers ungrammatical.[11] However, in many cases analyses can be found for inputs which exhausted default resource limits, simply by extending those resource limits. In this sense, it is possible to consider the ERG itself as a baseline contender for robust parsing: by assigning more generous resource limits, some previously unparseable inputs receive analyses. The quality of these analyses (as measured for instance by EDM precision) can in general be expected to be in line with the quality of other native ERG analyses, but their quantity (as measured by EDM recall) will be limited. The system referred to as *baseline* in the results below is the ERG with no extensions, but with resource limits of 8GB of RAM and unlimited time per item.

### 3.4.2 Bridging

This system, first described by Flickinger and Packard (2015), is one of two "self-help" methods, meaning it is implemented within the same formalism used by the rest of the grammar. The goal of the method is to allow the construction of a spanning analysis from analyses of shorter spans within an input that normally would not be able to combine with each other. For an input like the following, the ERG is normally unable to assemble the pieces into a coherent whole:

(1)   The dog, come quickly, the dog bit me!

Bridging allows an NP analysis of "The dog," an imperative analysis of "come quickly," and a declarative analysis of "the dog bit me!" to all be glued together into a single analysis that, while failing to say anything interesting about the relationships between those three parts, still makes the analyses of those parts accessible. This is implemented by two rules:

1. a unary rule promoting any ordinary analysis into a "bridge" analysis with the same span
2. a binary rule allowing the combination of two bridge analyses, where the lefthand daughter must itself be the unary bridging rule.

Although the system could in principle also be implemented as a single binary rule, the above design has the advantage of greatly reducing the cost of

---

[11]Though some applications of the ERG for language learning make use of an extension to the grammar consisting of a set of *mal-rules* described in Bender et al. (2004), which explicitly license certain ungrammatical structures.

parsing, since all of the results of the unary rule for a given span are combinatorially equivalent (i.e. they *pack*: see Oepen and Carroll (2000)), leading to just a single application of the binary rule per span, rather than one for each analysis of the left-hand side. Finally, to mitigate the possibility of lexical gaps not covered by unknown word handling, the implementation also posits a "bridge" lexical analysis of every input token.

The wealth of robust analyses spawned by the bridging mechanism gives rise to an important question: which one is the most useful? This problem of ambiguity is of course isomorphic to the problem of ambiguity experienced during ordinary parsing, and is amenable to the same solution, namely a statistical ranking model trained on a manually annotated treebank. Unfortunately, whereas the ERG comes with curated treebanks of roughly 100,000 trees falling within the scope of its ordinary grammatical rules, there are only very minimal curated treebanks of bridged analyses. As a result, the statistical model we use to select a parse contains no preferences regarding the bridging rules. With future research, it may be possible to significantly improve the performance of this disambiguation task.

Formally, bridging ensures that any input is analyzable. However, even with aggressive packing techniques, the sheer ambiguity presented by the bridging rules makes the unpacking problem quite challenging. In fact, the parser used in these experiments (namely ACE 0.9.24: `sweaglesw.org/linguistics/ace/`) was not able to solve the unpacking problem for a significant number of inputs, resulting in somewhat disappointing actual coverage.

### 3.4.3 Pacman

One clear disadvantage of the bridging approach is that once a bridged analysis has been invoked for a portion of an input sentence, any larger portion of the sentence containing that bridged element must also be bridged (to keep the approach at all tractable), and hence fails to produce informative semantics for the larger portions. So we explored a second "self-help" method, again adding robustness rules to the grammar, but instead of category-agnostic bridging rules, we provided rules that combine certain frequently appearing phrases such as NPs or VPs with an arbitrary word or phrase to their immediate left or right. The guiding intuition is that failure to parse a sentence often results

from some minor intrusion of an unexpected word or phrase, where if the parser could just skip over that obstacle, it could proceed to construct a quite serviceable derivation for the sentence. For example, the standard ERG will not succeed in analyzing the sentence *They might try to buy a cheap car, I suppose, or an old truck* because of the phrase *I suppose* appearing in the middle of a coordination of two NPs. But a rule that consumed *I suppose* along with the preceding NP *a cheap car* to produce an NP covering both phrases would then enable a usable analysis of the full sentence.

We term the use of such cheerfully greedy rules the "Pacman" approach, and for the present experiment, we limit ourselves to just two rules, one for building robust nominal phrases, and one for verb phrases; hence the shorthand label `pacnv`. The resulting grammar should be considerably more robust than the regular ERG, and possibly more successful in preserving meaning content than with bridging rules, but it cannot aspire to full coverage, since it crucially depends on finding a host nominal or verbal phrase next to the site of any parsing obstacle. We could expand the inventory of host phrase types to close this remaining coverage gap, but experiments to date indicate that increased ambiguity can quickly make parsing with a larger set of Pacman rules intractable.

### 3.4.4 PCFG approximation

Our third approach, referred to as `csaw` in the results below, is a straightforward reimplementation combining the methods of the `JigSaw` parser of Zhang and Krieger (2011) and the robust unification technique of Zhang et al. (In prep). The idea of combining these techniques belongs to the authors of those papers, and the details of how the systems work are given there as well. A summary will suffice to give readers the basic idea. First, a collection of derivation trees consistent with the underlying precision grammar is procured for use as training data. These trees can come either from gold standard treebanks or from parsing. The internal nodes of the trees are labeled with the rule name licensing the corresponding constituent (for example, the *head specifier rule*). Leaves correspond to lexemes, and are labeled with the name of the lexical type to which the lexeme belongs. In addition, some modifications are made to the shape of the tree: most lexical rules are collapsed onto the yield of the tree, and punctuation is split into binary structures. Finally, the trees can op-

tionally be decorated with grandparent information to a configurable depth. A PCFG is induced by maximum likelihood estimation from the decorated trees, with no smoothing, lexicalization, hierarchical splitting, or other enhancements (although such techniques would likely be fruitful avenues for future experiments).

The resulting PCFG is used in conjunction with a straightforward CKY parser to analyze arbitrary inputs. Depending on the level of grandparenting used and the size of the training set, the speed and coverage of this system can be high or low. For the purposes of this paper, two configurations are used: `csaw-tb`, which is an ungrandparented PCFG trained exclusively on hand-annotated gold-standard trees, and `csaw-ww`, which is a doubly-grandparented PCFG trained exclusively on automatically-disambiguated Wikipedia sentences. The two configurations are compared below:

| System | Non-terminals | Rules | Training set |
|---------|---------------|-------|--------------|
| csaw-tb | 236 | ∼36K | ∼100K |
| csaw-ww | 155042 | ∼5M | ∼50M |

Since the terminal symbols of these grammars are lexical types combined with sequences of lexical rules, these grammars are not directly useable for parsing plain text. Rather, they are dependent upon the preprocessing environment and lexical parsing capabilities of the ERG to prepare sequences or lattices of terminal symbols from which PCFG parsing can proceed.[12]

The decorations and shape modifications performed prior to inducing the PCFG are reverseable. Analyses found by the PCFG grammar can easily be converted[13] into trees that look superficially very similar to the derivation trees licensed by the precision grammar. In some cases, the sequence of rules and lexemes stipulated by these pseudo-derivations are in fact compatible with the unification constraints that define those rules and lexemes. Under these circumstances, the precision grammar can be used to "replay" the derivation and a coherent semantic analysis can be read out of the resulting feature structure, as in the case of ordinary parsing. Unfortunately, this can only happen when a fully consistent derivation *exists*, i.e. when the precision grammar was already

---

[12]This tight coupling facilitates semantic recovery, and is a major reason that more powerful "off-the-shelf" PCFG parsers were not used.

[13]Grandparent information must be removed, collapsed chains of lexical rules separated, and binary punctuation converted back into the ERG's lexical rule analysis.

capable of analyzing the input.

Zhang et al. (In prep) had the key insight that the unification constraints that these pseudo-derivations violate are usually in parts of the feature structure geometry that are isolated from the portion used for semantic composition. As a result, robust unification techniques such as the one they present, or older ones such as that of (Fouvry, 2003b), can be used to replay the pseudo-derivation and produce a feature structure that, while unreliable in terms of its syntactic properties, is largely consistent in terms of the semantic analysis it embodies. The csaw systems do just this, resulting in a (possibly somewhat incoherent) semantic analysis for nearly any input.[14]

### 3.4.5 A hybrid approach

The csaw approaches are "all-or-nothing", in the sense that if there is any portion at all of input that the precision grammar is unable to analyze, the entirety of the syntactic constraints must be relaxed in favor of the PCFG. This feels a bit like throwing the baby out with the bath water: the precision grammar may have a great deal of useful expertise to contribute to the analysis of large portions of the sentence, even if it is overly prescriptive or incomplete in its treatment of some particular detail. The hybrid-tb and hybrid-ww techniques attempt to exploit the best of both worlds. This is achieved by first allowing the corresponding csaw-based system to propose a single complete analysis, and then allowing the rules of the precision grammar to find additional ways to use the pieces of that robust analysis. Specifically, the constituents postulated by csaw are converted into chart edges and allowed to participate in chart parsing along with the rest of the edges licensed by the grammar. This allows a robust analysis of some particularly troublesome construction to combine with all other possible analyses of the rest of the sentence (including portions nested within the robust analysis). The best analysis is subsequently chosen by the statistical parse ranking model used when performing ordinary parsing with the precision grammar.

---

[14]Some inputs may be unparseable by the PCFG, of course. The robust semantic readout process can also fail in the empirically very rare case that robust unification produces a cyclic structure in an infelicitous location, such as the list of semantic predications. The latter occurred just once for us.

## 4 Results

Table 1 shows the coverage of each technique on each dataset. As expected, the baseline ERG system had the lowest coverage of any of the techniques for the alchemy45 and wsj00ab datasets. The pacnv system yielded even lower coverage than the baseline system for ws13r and semcor, although it showed promise on the alchemy45 where native coverage is very low. The reason that coverage for the bridging and pacnv systems is relatively low is that these techniques introduce very challenging ambiguity management problems for the parser, and as a result the parser frequently exhausts the resource allocation (which was the same as used for the baseline method). The PCFG-based methods all showed higher coverage than the ERG, across all datasets. The hybrid-ww method achieved the highest coverage on all datasets except semcor, where it was edged out by csaw-ww and hybrid-tb.

Table 2 shows parsing speeds. While the speeds may seem slow and the scores may seem low, it is important to bear in mind that these datasets are selected specifically to be hard to parse. As the parsing was performed on a cluster where CPU speed and load varied somewhat from experiment to experiment, the speed measurements are only comparable on a broad scale. Nonetheless, we clearly see that csaw-tb is by far the fastest system: its grammar is simpler, and the expense of unification-based parsing is entirely avoided.

The slowest systems are csaw-ww and hybrid-ww; the former because the corresponding PCFG is very large, and the latter because it includes the former as a component, and has to perform unification-based parsing as well. These patterns held across all four datasets.

Table 2 also shows EDM F1 scores, a fine-grained evaluation of the semantic dependencies encoded in the MRSes recovered by each parser. Three datasets pattern the same: ws13r, semcor, and wsj00ab. On these datasets, hybrid-ww produced the highest scores, while bridging, pacnv, and csaw-tb earned the lowest marks, edged out even by baseline.

On alchemy45, the story is different: baseline finds no competitors for the lowest position, while csaw-ww performs slightly better than hybrid-ww, taking the top score. The highly efficient csaw-tb also performs better on

| Method | alchemy45 | ws13r | semcor | wsj00ab |
|---|---|---|---|---|
| baseline | 17.78% | 61.35% | 88.74% | 84.21% |
| bridging | 66.67% | 74.40% | 90.91% | 90.79% |
| pacnv+ut | 53.33% | 54.11% | 87.45% | 85.53% |
| csaw-tb | 97.78% | 77.29% | 98.27% | 98.68% |
| csaw-ww | **100.00%** | 83.09% | **100.00%** | **100.00%** |
| hybrid-tb | 97.78% | 88.41% | **100.00%** | 98.68% |
| hybrid-ww | **100.00%** | **90.82%** | 99.13% | **100.00%** |

Table 1: Coverage for each technique.

| Method | alchemy45 | | ws13r | | semcor | | wsj00ab | |
|---|---|---|---|---|---|---|---|---|
| | F1 | T | F1 | T | F1 | T | F1 | T |
| baseline | 28.83 | 44.1 | 52.28 | 31.4 | 79.60 | 7.7 | 72.78 | 10.6 |
| bridging | 42.07 | 41.2 | 44.31 | 48.3 | 69.61 | 22.4 | 66.11 | 21.9 |
| pacnv+ut | 43.85 | 29.0 | 42.27 | 89.7 | 72.70 | 16.6 | 66.16 | 27.6 |
| csaw-tb | 68.51 | **2.1** | 48.87 | **1.0** | 67.81 | **0.6** | 67.26 | **0.7** |
| csaw-ww | **77.11** | 247.0 | 60.98 | 224.3 | 78.51 | 135.8 | 74.48 | 132.2 |
| hybrid-tb | 69.76 | 39.2 | 63.73 | 20.4 | 78.76 | 13.4 | 76.17 | 14.1 |
| hybrid-ww | 75.56 | 219.2 | **68.47** | 228.2 | **81.52** | 122.0 | **78.61** | 119.8 |

Table 2: EDM F1 and wallclock seconds per item for each technique.

this dataset than it did on the others.

## 5 Discussion

The dichotomy between performance on `alchemy45` and the other three datasets is interesting. There are two things about that dataset that are unique: first, it is the only dataset of the four that was annotated against the `1214` version of the ERG instead of against the `trunk` version. Second, it has by far the lowest proportion of sentences that the ERG can analyze unassisted. From that perspective, it is the best indicator of performance on inputs where robust techniques are needed the most. However, it is also the smallest of the four datasets, so results may not be as reliable.

One clear result is that the PCFG-based systems are capable of greatly enhancing the coverage of the ERG, while producing analyses whose quality is better than the other robust systems evaluated. While the speed of the best-performing `csaw-ww` and `hybrid-ww` methods is somewhat uninspiring, we console ourselves by noting that the method is at least tractable, and the performance is probably acceptable for some applications where coverage and accuracy are at a premium while time is not. When the utmost accuracy is not required for robust parses, the `hybrid-tb` system is not noticeably slower than

the baseline, yet has a much higher coverage, with scores somewhat lower than `hybrid-ww` but still much better than `baseline` for the `alchemy45` dataset. The fact that the overall EDM F1 score for the `hybrid-tb` method was slightly lower than the very respectable `baseline` score on the `semcor` dataset indicates that successful deployment of this technique may be predicated upon restricting its use to cases where the ERG in isolation has already failed to produce an analysis.

In the present work, the parse ranking model is not adapted to the novel situations in which it is used, and hence likely does not perform as well as it could. New techniques are needed to train the ranking model to make informed decisions about these situations. Although coverage would be unaffected, accuracy could potentially be improved. Also, evaluating the difference in performance on data strictly outside of the coverage of the underlying precision grammar compared to data in scope would be profitable, as would exploring techniques for managing the decision about when to deploy robustness.

While it remains to be seen whether the promising coverage gains and accuracy figures reported here can be borne out in extrinsic evaluation, we believe there is reason for optimism, as downstream applications may be able to mitigate coverage issues using these techniques.

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
