# Peer review of "A Comparison of Robust Parsing Methods for HPSG"

_ACL 2017 — decision unknown_

[Official Review · Reviewer 1 · rating 3 · confidence 4]
soundness 3 · originality 3 · clarity 5 · impact 3 · substance 4 · appropriateness 5 · meaningful comparison 3 · presentation format Oral Presentation

- Strengths:
 * Elaborate evaluation data creation and evaluation scheme.
 * Range of compared techniques: baseline/simple/complex

- Weaknesses:
 * No in-depth analysis beyond overall evaluation results.

- General Discussion:
This paper compares several techniques for robust HPSG parsing.

Since the main contribution of the paper is not a novel parsing technique but
the empirical evaluation, I would like to see a more in-depth analysis of the
results summarized in Table 1 and 2.
It would be nice to show some representative example sentences and sketches of
its analyses, on which the compared methods behaved differently.

Please add EDM precision and recall figures to Table 2.
The EDM F1 score is a result of a mixed effects of (overall and partial)
coverage, parse ranking, efficiency of search, etc.
The overall coverage figures in Table 1 are helpful but addition of EDM recall
to Table 2 would make the situations clearer.

Minor comment:
- Is 'pacnv+ut' in Table 1 and 2 the same as 'pacnv' described in 3.4.3?

[Official Review · Reviewer 2 · rating 2 · confidence 4]
soundness 3 · originality 3 · clarity 4 · impact 3 · substance 3 · appropriateness 5 · meaningful comparison 3 · presentation format Poster

- Strengths:

Well-written.

- Weaknesses:

Although the title and abstract of the paper suggest that robust parsing
methods for HPSG are being compared, the actual comparison is limited to only a
few techniques applied to a single grammar, the ERG (where in the past the 
choice has been made to create a treebank for only those sentences that are in
the coverage of the grammar). Since the ERG is quite idiosyncratic in this
respect, I fear that the paper is not interesting for researchers working in
other precision grammar frameworks.

The paper lacks comparison with robustness techniques that are routinely
applied for systems based on other precision grammars such as various systems
based on CCG, LFG, the Alpage system for French, Alpino for Dutch and there is
probably more. In the same spirit, there is a reference for supertagging to
Dridan 2013 which is about supertagging for ERG whereas supertagging for other
precision grammar systems has been proposed at least a decade earlier.

The paper lacks enough detail to make the results replicable. Not only are
various details not spelled out (e.g. what are those limits on resource
allocation), but perhaps more importantly, for some of the techniques that are
being compared (eg the robust unification), and for the actual evaluation
metric, the paper refers to another paper that is still in preparation.

The actual results of the various techniques are somewhat disappointing. With
the exception of the csaw-tb method, the resulting parsing speed is extreme -
sometimes much slower than the baseline method - where the baseline method is a
method in which the standard resource limitations do not apply. The csaw-tb
method is faster but not very accurate, and in any case it is not a method
introduced in this paper but an existing PCFG approximation technique.

It would be (more) interesting to have an idea of the results on a
representative dataset (consisting of both sentences that are in the coverage
of the grammar and those that are not). In that case, a comparison with the
"real" baseline system (ERG with standard settings) could be obtained.

Methodological issue: the datasets semcor and wsj00ab consist of sentences
which an older version of ERG could not parse, but a newer version could. For
this reason, the problems in these two datasets are clearly very much biased.
It is no suprise therefore that the various techniques obtain much better
results on those datasets. But to this reviewer, those results are somewhat
meaningless. 

minor:

EDM is used before explained

"reverseability"

- General Discussion:

[Official Review · Reviewer 3 · rating 3 · confidence 4]
soundness 3 · originality 3 · clarity 4 · impact 3 · substance 4 · appropriateness 5 · meaningful comparison 3 · presentation format Oral Presentation

- Strengths:

- technique for creating dataset for evaluation of out-of-coverage items, that
could possibly be used to evaluation other grammars as well. 
- the writing in this paper is engaging, and clear (a pleasant surprise, as
compared to the typical ACL publication.)

- Weaknesses:
- The evaluation datasets used are small and hence results are not very
convincing (particularly wrt to the alchemy45 dataset on which the best results
have been obtained)
- It is disappointing to see only F1 scores and coverage scores, but virtually
no deeper analysis of the results. For instance, a breakdown by type of
error/type of grammatical construction would be interesting. 
- it is still not clear to this reviewer what is the proportion of out of
coverage items due to various factors (running out of resources,  lack of
coverage for "genuine" grammatical constructions in the long tail, lack of
coverage due to extra-grammatical factors like interjections, disfluencies,
lack of lexical coverage, etc. 

- General Discussion:

This paper address the problem of "robustness" or lack of coverage for a
hand-written HPSG grammar (English Resource Grammar). The paper compares
several approaches for increasing coverage, and also presents two creative ways
of obtaining evaluation datasets (a non-trivial issue due to the fact that gold
standard evaluation data is by definition available only for in-coverage
inputs). 

Although hand-written precision grammars have been very much out of fashion
for a long time now and have been superseded by statistical treebank-based
grammars, it is important to continue research on these in my opinion. The
advantages of high precision and deep semantic analysis provided by these
grammars has not been
reproduced by non-handwritten grammars as yet. For this reason, I am giving
this paper a score of 4, despite the shortcomings mentioned above.